# Use of Closed Incision Negative Pressure Therapy (ciNPT) in Breast Reconstruction Abdominal Free Flap Donor Sites

**DOI:** 10.3390/jcm10215176

**Published:** 2021-11-05

**Authors:** Jennifer Wang, Zyg Chapman, Emma Cole, Satomi Koide, Eldon Mah, Simon Overstall, Dean Trotter

**Affiliations:** 1Department of Plastic and Reconstructive Surgery, Royal Melbourne Hospital, Melbourne, VIC 3050, Australia; emma.j.cole@mh.org.au (E.C.); satomi.koide@mh.org.au (S.K.); eldonmah@gmail.com (E.M.); simonoverstall@gmail.com (S.O.); deantrotter@yahoo.com (D.T.); 2Faculty of Medicine, Dentistry and Health Sciences, University of Melbourne, Melbourne, VIC 3053, Australia; zchapman@student.unimelb.edu.au

**Keywords:** breast reconstruction, DIEP, wound management, negative pressure dressing, Prevena

## Abstract

Background: Closed incision negative pressure therapy (ciNPT) may reduce the rate of wound complications and promote healing of the incisional site. We report our experience with this dressing in breast reconstruction patients with abdominal free flap donor sites. Methods: A retrospective cohort study was conducted of all patients who underwent breast reconstruction using abdominal free flaps (DIEP, MS-TRAM) at a single institution (Royal Melbourne Hospital, Victoria) between 2016 and 2021. Results: 126 female patients (mean age: 50 ± 10 years) were analysed, with 41 and 85 patients in the ciNPT (Prevena) and non-ciNPT (Comfeel) groups, respectively. There were reduced wound complications in almost all outcomes measured in the ciNPT group compared with the non-ciNPT group; however, none reached statistical significance. The ciNPT group demonstrated a lower prevalence of surgical site infections (9.8% vs. 11.8%), wound dehiscence (4.9% vs. 12.9%), wound necrosis (0% vs. 2.4%), and major complication requiring readmission (2.4% vs. 7.1%). Conclusion: The use of ciNPT for abdominal donor sites in breast reconstruction patients with risk factors for poor wound healing may reduce wound complications compared with standard adhesive dressings; however, large scale, randomised controlled trials are needed to confirm these observations. Investigation of the impact of ciNPT patients in comparison with conventional dressings, in cohorts with equivocal risk profiles, remains a focus for future research.

## 1. Introduction

The abdominal free flap is considered the gold standard in most cases of breast reconstruction but may be associated with significant wound complications. Complications such as surgical site infections (SSI), wound dehiscence and tissue necrosis contribute to morbidity, health care costs and are burdensome for patients and surgeons. Closed incision negative pressure therapy (ciNPT) has been reported to reduce the incidence of wound complications by maintaining a closed wound environment, promoting perfusion and eliminating exudate via constant negative pressure [1,2,3].

Growing research into the wound healing process at a molecular level has led to continual advances in wound care. Morykwas et al. first demonstrated in 1997 that the application of sub-atmospheric pressure up to 125 mmHg onto wounds increased local tissue perfusion using animal and scientific studies [3]. Creating a suction force allows the drainage of excess interstitial fluid, reducing both physical and chemical deterrents of wound healing. Whilst this is not a new concept, its clinical success over the years has encouraged clinicians to trial the same technique to closed incisional wounds. Furthermore, extensive research performed by Ogawa et al. has demonstrated that minimizing skin tension also plays a critical role in the repair and regenerative wound response, adding to the theoretical advantage of ciNPT [4].

Whilst several studies have shown promising results with ciNPT, there is limited research on its effectiveness for abdominal free flap donor sites. For instance, Zwanenburg et al. [5] conducted a meta-analysis and meta-regression and showed a beneficial reduction in surgical site infections with ciNPT. A more recent Cochrane review of negative pressure wound therapy for all surgical wounds concluded that ciNPT dressings may reduce rates of surgical site infections (moderate-level evidence); however, there was insufficient evidence to allow any recommendation regarding its effect on wound dehiscence or risk of death [6]. Findings of other studies have shown inconsistent results and mixed reviews of negative wound therapy [6,7]. Thus, further research is warranted to evaluate the effectiveness of negative pressure wound therapy in clinical settings.

Here, we assessed whether donor site wound management with ciNPT is associated with reduced wound complications in breast reconstruction patients with abdominal free flaps compared with standard adhesive wound care.

## 2. Materials and Methods

After local ethical review board approval by Melbourne Health (ID: QA2021007), a retrospective cohort study was conducted of patients who underwent breast reconstruction surgery using abdominal free flaps (DIEP, MS-TRAM) at The Royal Melbourne Hospital over a 5-year period, between October 2016 and April 2021. Standard wound care (non-ciNPT) for abdominal free flap donor sites involves the use of a hydrocolloid dressing (Comfeel^®^ manufactured by Coloplast, Melbourne, VIC, Australia). The incorporation of ciNPT at our centre for the management of breast reconstruction abdominal wounds began in 2016. This utilises the application of a self-adhesive foam-based dressing over the incision site, connected to a suction pump that applies continuous negative pressure at 125 mmHg (PREVENA^™^ Incision Management System manufactured by KCI USA, INC., 12930 W Interstate 10, San Antonio, TX, USA). Patients who were considered to be at ‘high risk’ of poor wound healing by the surgical team, i.e., those with diabetes and/or obesity, were preferentially selected to trial the ciNPT dressing.

All surgeries were undertaken at a single institution by the same surgical team. All patients received the same perioperative care with intraoperative antibiotics, and postoperative prophylactic anticoagulation (enoxaparin or heparin). All ciNPT dressings remained in place for 5–7 days (see Figure 1). The primary outcome was surgical site infection. Secondary outcomes included wound dehiscence, haematoma, seroma, major complication (i.e., complication requiring readmission) and unplanned return to theatre. Patients were followed up weekly for the first 3 postoperative weeks, then at 6 weeks. No patients were lost to follow up.

Data was collected from medical records and recorded using a REDcap database. Statistical analysis was conducted using Stata (Release 17; Statacorp; College Station, TX, USA) for both descriptive and comparative testing. Ordinal variables were compared using chi-squared or Fisher’s exact analysis involving frequencies less than five. Continuous variables were compared using non-parametric tests (Mann–Whitney U test) due to the skewed distribution. Statistical significance was set as *p* < 0.05 a priori.

## 3. Results

### 3.1. Patient Characteristics

One hundred and twenty-six female patients were included in this study. In a non-randomised fashion, forty-one received ciNPT and eighty-five received non-ciNPT. Patient demographics are summarised in Table 1. The two groups differed significantly in terms of comorbidities. Patients with diabetes and obesity (BMI > 30 g/m^2^) were significantly more likely to receive ciNPT (*p* = 0.005 and *p* = 0.002, respectively). There was no other statistical difference in characteristics between treatment groups (*p* > 0.05).

### 3.2. Operative Characteristics

Operative characteristics are summarized in Table 2. Bilateral and unilateral mastectomies were equally represented in this population. No significant difference was identified in the flap reconstruction choice between cohorts. All patients underwent breast reconstruction with either a deep inferior epigastric perforator (DIEP) free flap only, or combination of DIEP and muscle-sparing transverse rectus abdominis myocutaneous (MS-TRAM) flaps. The median length of surgery was comparable between the groups.

### 3.3. Postoperative Outcomes

There was a non-significant reduced prevalence of wound complications in almost all outcomes measured in the ciNPT group compared with the non-ciNPT group (see Table 3). The ciNPT group demonstrated a lower rate of surgical site infections (9.8% vs. 11.8%; *p* = 0.737), wound dehiscence (4.9% vs. 12.9%; *p* = 0.247), wound necrosis (0% vs. 2.4%; *p* = 1.0), total number of patients with complications (17.1% vs. 25.9%; *p* = 0.271) and major complication requiring readmission (2.4% vs. 7.1%; *p* = 0.646). Only the proportion of seroma was marginally higher in the ciNPT group (4.9% vs. 1.2%; *p* = 0.247). Length of hospital stay was statistically longer in the ciNPT group (6 days vs. 5 days; *p* = 0.009). In both groups, 2.4% of patients had a wound complication requiring an unexpected return to theatre. In all of these cases, patients had known risk factors for poor wound healing.

## 4. Discussion

We investigated whether donor site wound management with ciNPT was associated with reduced wound complications in breast reconstruction patients with abdominal free flaps compared with standard adhesive wound care. Findings showed a reduced prevalence in surgical site infections (9.8% vs. 11.8%), wound dehiscence (4.9% vs. 12.9%) and several other complications; however, these did not reach statistical significance, likely due to our small sample size. There was a significantly higher proportion of patients with diabetes and obesity in the ciNPT group (*p* < 0.05). Despite the disproportionate allocation of high-risk patients to the ciNPT group, this study demonstrated a non-inferiority in wound healing across multiple domains when comparing the two cohorts. This was the case for total wound complications (17.1% vs. 25.91%), wound necrosis (0% vs. 2.4%), and major complications requiring readmission (2.4% vs. 7.1%). The ciNPT cohort included two patients that developed seromas, compared with one reported seroma case in the non-ciNPT group; however, this was not statistically significant. The prevalence of patients requiring return to theatre was equivalent for both cohorts (2.4%). 

Our study findings add to the body of literature on this topic. Several studies have shown promising results with ciNPT for general wound management [8,9,10,11,12,13,14,15,16]. Smolle et al. demonstrated a reduction in wound complications in patients who received ciNPT for orthopaedic surgical sites (total joint arthroplasty) and colorectal laparotomy sites [14]. A randomised control trial conducted by Tanaydin et al. compared ciNPT with standard adhesive dressings (fixation strips) in breast reduction mammoplasty surgical sites and found that ciNPT was associated with reduced wound complications (*p* < 0.04) and an improvement in scar quality [17]. A 2020 Cochrane Review of negative pressure wound therapy for all surgical wounds concluded that ciNPT dressings may reduce rates of surgical site infections (moderate-level evidence); however, there was insufficient evidence to allow any recommendation regarding its effect on wound dehiscence or risk of death [6]. 

Whilst there are several studies investigating the use of ciNPT, very limited research exists on its effectiveness in reconstructive breast surgery. This is the largest study to our knowledge investigating the use of ciNPT for breast reconstruction abdominal donor sites. Fang et al. retrospectively compared outcomes in 10 patients with abdominal free flap breast reconstructions (ciNPT = 5, non-ciNPT = 5), concluding that ciNPT dressings resulted in faster wound healing and better cosmesis; however, without statistical significance [16]. Muller-Sloof et al. conducted a prospective randomised control trial of 51 patients (ciNPT = 25, non-ciNPT = 26) who underwent breast reconstruction with abdominal free flaps, comparing ciNPT (Prevena) with standard adhesive dressings, demonstrating a reduction in incidence of wound dehiscence in the ciNPT cohort [15]. Our study, benefiting from a larger sample size than previous studies, adds to an existing body suggesting a potential superiority of ciNPT compared with conventional adhesive dressings. Research with large, randomised-controlled trials needs to be conducted to further assess the wound outcomes in this cohort.

The results from this study demonstrate a non-inferiority in wound outcomes for patients at high risk of poor wound healing. There was intentional surgeon selection of ciNPT for patients with known risk factors for poor wound healing, i.e., diabetes and obesity, as evidenced by the statistically significant comorbidity differences between the two groups. The association between diabetes and obesity with poor wound outcomes following DIEP surgeries has been previously well documented [5,6,17,18,19]. Negative pressure dressings have been identified as superior in certain abdominal operations for higher risk patients [7,8,9,10] and their use is being trialed in more abdominal surgeries such as donor site closure as in our study population. The use of ciNPT showed improved outcomes to a level proportionate to patients without raised risk profiles, although without statistical significance. One can justify employing negative pressure dressings on patients at high risk of poor wound healing, such as in diabetes and obesity, which is now further being implemented at our centre.

Whilst there was a statistical difference in length of hospital stay between the two groups, this was not deemed a relevant outcome measure of wound healing in our study. Patients who received ciNPT were required to remain an inpatient for at least 5–7 days for nursing care until the Prevena dressing was removed, precluding this measure from being clinically relevant when measuring wound complication outcomes. Patients with comorbidities such as diabetes may require increased length of stay to optimize glycaemic management pre- and post-surgery [19,20,21]. In addition, obesity has been linked to an increased length of stay with longer observation periods often due to a prolonged, more complex postoperative recovery [18,21,22,23,24].

There are several limitations to this study. This was a non-randomised study, with evidence of selection bias reflected in the skewed patient population. The retrospective nature of the study lends itself to potential information bias, as incomplete records were excluded. The sample size, while larger than other similar studies in the current medical literature, remains small. Although it seems that ciNPT dressing systems improve wound complications for higher-risk patients, conclusions cannot be drawn regarding whether this also improves outcomes for low-risk patients undergoing breast reconstruction with abdominal free flap donor sites. Further investigation with prospective, randomized cohorts of a larger sample size would better elucidate this.

## 5. Conclusions

The use of ciNPT for abdominal donor sites in breast reconstruction patients with risk factors for poor wound healing reduces wound complications compared with standard adhesive dressings. We therefore recommend the use of ciNPT for patients with high-risk of wound complications, such as those with diabetes and obesity. The benefit of these dressings compared with conventional dressings has yet to be demonstrated for patients with standard risk profiles. Further investigation with a prospective randomised study will better elucidate the wound outcomes in this patient cohort and validate the use of ciNPT.

## Figures and Tables

**Figure 1 jcm-10-05176-f001:**
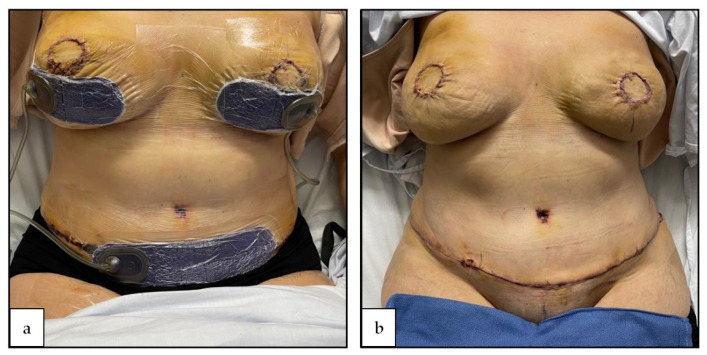
(**a**) Application of ciNPT dressings to abdominal donor site (and recipient sites) (**b**) ciNPT dressings removed on post-operative day 5.

**Table 1 jcm-10-05176-t001:** Preoperative characteristics of ciNPT (Prevena) vs. non-ciNPT (Comfeel) groups.

Variable	Total Sample (*n* = 126)	ciNPT (*n* = 41)	Non-ciNPT (*n* = 85)	*p* Value
Age (years; mean ± SD)	50.0 ± 10	50.5 ± 24	49.7 ± 11	0.676
Non-smoker, *n* (%)	94 (74.6)	29 (70.7)	65 (76.5)	0.517
Ex-smoker, *n* (%)	25 (19.8)	8 (19.5)	17 (20.0)	0.808
Current smoker, *n* (%)	7 (5.6)	4 (9.8)	3 (3.5)	0.393
Diabetes mellitus, *n* (%)	7 (5.6)	6 (14.6)	1 (1.2)	0.005 *
Median BMI (kg/m^2^)	27.3	31	26.6	0.124
BMI ≥ 30 kg/m^2^, *n* (%)	43 (34.1)	22 (53.7)	21 (24.7)	0.002 *
Previous chemotherapy, *n* (%)	33 (26.2)	13 (31.7)	20 (23.5)	0.328
Neoadjuvant chemotherapy, *n* (%)	27 (21.4)	11 (26.8)	16 (18.8)	0.305
Anticoagulation, *n* (%)	4 (3.2)	3 (7.3)	1 (1.2)	0.101

* Significant *p* values. BMI = body mass index. SD = standard deviation.

**Table 2 jcm-10-05176-t002:** Operative characteristics: ciNPT vs. non-ciNPT patients.

Operative Characteristic	Total Sample (*n* = 126)	ciNPT (*n* = 41)	Non-ciNPT (*n* = 85)	*p* Value
Type of Surgery				
Bilateral mastectomy, *n* (%)	63 (50.6)	18 (43.9)	45 (52.9)	0.447
Unilateral mastectomy, *n* (%)	63 (50.0)	23 (65.1)	40 (47.1)	0.342
Type of Flap				
DIEP, *n* (%)	120 (95.2)	39 (95.1)	81 (95.3)	1.000
MS-TRAM, *n* (%)	0	0	0	-
Both DIEP and MS-TRAM, *n* (%)	6 (4.8)	2 (4.9)	4 (4.7)	0.966
Length of surgery in hours; median	6.5	6.5	6.5	0.765

DIEP = deep inferior epigastric perforator. MS-TRAM = muscle-sparing transverse rectus abdominis myocutaneous.

**Table 3 jcm-10-05176-t003:** Postoperative outcomes: ciNPT vs. non-ciNPT patients.

Wound Complication	Total Sample (*n* = 126)	ciNPT (*n* = 41)	Non-ciNPT (*n* = 85)	*p* Value
Total wound complications, *n* (%)	42 (33.3)	10 (23.9)	32 (37.6)	0.162
Number of patients with wound complication(s), *n* (%)	29 (23.0)	7 (17.1)	22 (25.9)	0.271
Surgical site infection, *n* (%)	14 (11.1)	4 (9.8)	10 (11.8)	0.737
Wound dehiscence, *n* (%)	13 (10.3)	2 (4.9)	11 (12.9)	0.219
Wound necrosis, *n* (%)	2 (1.6)	0	2 (2.4)	1.000
Seroma, *n* (%)	3 (2.4)	2 (4.9)	1 (1.2)	0.247
Major complication requiring readmission, *n* (%)	7 (5.5)	1 (2.4)	6 (7.0)	0.646
Return to theatre, *n* (%)	3 (2.4)	1 (2.4)	2 (2.4)	0.429
Length of hospital stay in days (median, range)	5 (5–22)	6 (5–14)	5 (5–22)	0.009

## Data Availability

The data presented in this study are available on request from the corresponding author. The data are not publicly available due to consideration for patients’ privacy.

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
