# Peer review of "Use of Closed Incision Negative Pressure Therapy (ciNPT) in Breast Reconstruction Abdominal Free Flap Donor Sites"

_jcm, 2021, doi:10.3390/jcm10215176_

Round 1

Reviewer 1 Report

The authors present the use of ciNPT in breast reconstruction abdominal free flap donor sites. The authors concluded that the use of ciNPT in patients with risk factors for poor wound healing may reduce wound complications compared to standard adhesive dressings; however, large-scale, randomized
controlled trials are needed to confirm these observations. Investigation of the impact of ciNPT patients in comparison to conventional dressings, in cohorts with equivocal risk profiles, remains a focus for future research.

Minor comments:

Abstract instead of „this” these observations should be used.

Page 2 woud should be corrected to wound, such as warrated to warranted and evauate to evaluate.

In the methodological section „Data from medical records was entered using a REDcap database.” was should be changed to were.

In discussion, potenital should be corrected to potential and futher to further.

Major Comments:

In the Materials and Methods section, the selection should be described better. Of course that it is described later but still disturbing that it is not presented there. 

There were no significant differences in complications, but this was a retrospective non-randomized study, with evidence of selection bias.  In my opinion, the authors would found significant differences using randomized selection. 

The 2 vs 1 seroma are not comparable events.

Completely agree that further investigation with prospective, randomized cohorts of a larger sample size would better present the differences.

The method and the patient selection, randomization should have been better designed to get significant differences.

The patient hospitalization is one day longer in ciNPT group that can be addressed to the comorbidities. Unfortunately, it decreases medical costs.

Prevena is also an expensive device that can decreases expenses as well.

Further studies should compare de cost- effectiveness of this method as well.

Author Response

Response to Reviewer 1 Comments

Minor comments:

  1. Abstract instead of “this” these observations should be used.

Response 1: Thanks, this has been corrected.

  1. Page 2 woud should be corrected to wound, such as warrated to warranted and evauate to evaluate.

Response 2: These have been corrected.

  1. In the methodological section “Data from medical records was entered using a REDcap database.” was should be changed to were.

Response 3: Thank you, we have updated this sentence.

  1. In discussion, potenital should be corrected to potential and futher to further.

Response 4: These have also been corrected.

Major Comments:

  1. In the Materials and Methods section, the selection should be described better. Of course that it is described later but still disturbing that it is not presented there. 

Response 5: This has been addressed. Patients deemed at high risk of poor wound healing (diabetes, obesity) were preferentially selected by the surgical team to receive the ciNPT dressing.

  1. There were no significant differences in complications, but this was a retrospective non-randomized study, with evidence of selection bias. In my opinion, the authors would’ve found significant differences using randomized selection. 

Response 6: yes, we agree, and hope that study will be the impetus for a RCT.

  1. The 2 vs 1 seroma are not comparable events.

Response 7: agreed, not clinically or statistically significant.

  1. Completely agree that further investigation with prospective, randomized cohorts of a larger sample size would better present the differences.

Response 8: agree.

  1. The method and the patient selection, randomization should have been better designed to get significant differences.

Response 9: As it was a retrospective study, randomization was not possible.

  1. The patient hospitalization is one day longer in ciNPT group that can be addressed to the comorbidities. Unfortunately, it decreases medical costs. Prevena is also an expensive device that can decrease expenses as well.

Response 10:  We have addressed this since the study. Now patients are discharged the same time as non ciNPT patients. Our expectation is that the extra day in hospital with a prevena will ultimately reduce costs due to complications. Further study into cost effectiveness will hopefully clarify this hypothesis.

  1. Further studies should compare the cost-effectiveness of this method as well.

Response 11:  absolutely agree - we hope to do this.

Reviewer 2 Report

The following issues should be addressed with a reply point-by-point

The Syntax is clumsy. I recommend extensive revision to improve readability

“Page 2 Patients who were con-sidered ‘high risk’ of wound complication by the surgical team were selected to trial the ciNPT dressing.”

Later in the text, the authors stated that the selection bias was done on purpose. In my opinion selection bias is a strong limit in an outcomes study.

Page 4

"Our study findings add to the body of literature on this topic. Indeed, whilst several studies have shown promising results with ciNPT, [7–16] limited research exists on its ef-fectiveness in abdominal free flap donor sites."

How abdominal free flap donor site is different than the other wounds that were treated with ciNPT in the cited studies? this should be clarified

Page 4

“Muller-Sloof et al. conducted a prospective randomised control trial of 51 patients who underwent breast reconstruction with abdominal free flaps, comparing the Prevena dressing with standard adhesive dressing, with a statistically significant reduction in inci-dence of wound dehiscence in the Prevena cohort [14]. Fang et al. compared outcomes in 10 patients with abdominal free flap breast reconstructions, concluding that ciNPT dressings resulted in faster wound healing and better cosmesis, however without statistical signifi-cance [15]. Our findings strengthen the potenital benefits of ciNPT in this clinical cohort, which needs to be futher investigated in large, randomised controlled trials”

The authors should clarify how a retrospective non randomized study strengthened the findings of study “14 “and “15” or added new findings to the literature

What is novel in the present manuscript should be clarified

Author Response

Response to Reviewer 2 Comments

  1. The Syntax is clumsy. I recommend extensive revision to improve readability 

Response 1: Thanks. This has been corrected.

  1. Page 2: Patients who were considered ‘high risk’ of wound complication by the surgical team were selected to trial the ciNPT dressing. Later in the text, the authors stated that the selection bias was done on purpose. In my opinion selection bias is a strong limit in an outcomes study.

Response 2: Agree. If there hadn’t been selection bias, the results may have been different. This has been acknowledged in the paper, with the need now for a randomised, prospective trial to further assess the wound outcomes in this cohort.

  1. Page 4: "Our study findings add to the body of literature on this topic. Indeed, whilst several studies have shown promising results with ciNPT, [7–16] limited research exists on its effectiveness in abdominal free flap donor sites." How abdominal free flap donor site is different than the other wounds that were treated with ciNPT in the cited studies? this should be clarified.

Response 3: This has been clarified in the discussion. Current studies support use of ciNPT in orthopaedic sites (total joint arthroplasty), general surgery (laparotomy sites, incisional hernia repair sites), breast reduction mammoplasty sites. However there is limited evidence exploring its use for abdominal donor sites in breast reconstruction patients.

  1. Page 4: “Muller-Sloof et al. conducted a prospective randomised control trial of 51 patients who underwent breast reconstruction with abdominal free flaps, comparing the Prevena dressing with standard adhesive dressing, with a statistically significant reduction in inci-dence of wound dehiscence in the Prevena cohort [14]. Fang et al. compared outcomes in 10 patients with abdominal free flap breast reconstructions, concluding that ciNPT dressings resulted in faster wound healing and better cosmesis, however without statistical signifi-cance [15]. Our findings strengthen the potential benefits of ciNPT in this clinical cohort, which needs to be further investigated in large, randomised controlled trials”

The authors should clarify how a retrospective non-randomized study strengthened the findings of study “14 “and “15” or added new findings to the literature.

Response 4: There is very limited research specifically looking at ciNPT use in breast reconstruction abdominal free flap donor sites. Our study supports the findings from the two published studies, benefiting from a larger sample size than Muller-Sloof [51 patients (ciNPT = 25, non-ciNPT = 26)] and Fang [10 patients (ciNPT = 5, non-ciNPT = 5)]. Whilst these two studies have matched cohorts with similar comorbidities, our study adds to the literature by also demonstrating a non-inferiority in wound healing in high risk patients (diabetic, obese).

  1. What is novel in the present manuscript should be clarified

Response 5: Whilst we haven’t claimed to be presenting anything ground-breaking, our study does add to the limited body of research on ciNPT use for abdominal free flap donor sites. We also demonstrate a non-inferiority in wound healing amongst high risk patients. Of course, we acknowledge that there are several limitations to our study, and we plan to conduct prospective randomised trials in the future at our institution.

Round 2

Reviewer 2 Report

Dear Authors, i am aware that you put much effort in drawing and presenting your study. However, in my opinion, an outcome study should present novel results for a specific patient population or statistically significant results that could lead future decisions of clinicians/surgeons. Unfortunately, in your study i did not find either of them.

Author Response

Thank you for the feedback. We would first like to comment on the English language and style. The manuscript was written by a native English speaker. It was also edited and revised by multiple native English-speaking medical professionals. Google spell checker has been applied and the grammar has been checked multiple times by native English speakers.

There is very limited research on the use of ciNPT on breast reconstruction abdominal free flap sites (only two currently published papers), both stating that further research needs to be performed to elucidate the true benefit of ciNPT in this context. This paper not only adds to the body of research in this specific field but is the biggest series to date. Whilst we appreciate there are no statistically significant results, this is because the groups were heterogenous and the study was performed retrospectively. Despite this, there are strong trends towards a potential benefit in ciNPT, and until we can perform a large RCT, it is very unlikely that statistical significance will be demonstrated.

We firmly believe this paper does significantly contribute to the literature and is already changing our practice. We are now utilising ciNPT more often both for high-risk patients (diabetes, obesity) and in general.